# Increasing the Compressive Strength of Helicoidal Laminates after Low-Velocity Impact upon Mixing with 0° Orientation Plies and Its Analysis

**DOI:** 10.3390/ma16134599

**Published:** 2023-06-26

**Authors:** Zhefeng Yu, Xin Du, Rui Liu, Qiwu Xie, Xiaojing Zhang, Qiao Zhu

**Affiliations:** Aerospace Structure Research Center, School of Aeronautics and Astronautics, Shanghai Jiao Tong University, Shanghai 200240, China

**Keywords:** helicoidal composite, low velocity impact, compressive strength, laminate design

## Abstract

The helicoidal laminate is a kind of nature inspired fiber reinforced polymer, and the ply orientation affects their mechanical properties for engineering structural applications. A variety of helicoidal laminates with uniform and non-linear pitch angles mixed with additional 0° plies are fabricated to investigate the impact resistance through low-velocity impact and after-impact compression tests. Additionally, helicoidal laminates with constant pitch angles, quasi-isotropic laminate, and cross-ply laminates are also fabricated for a comparative study. The impact characteristics and the compressive strength are analyzed in view of the impact model, shear stress distribution, and fracture toughness. The results suggest that 10° or 20° are the better basic pitch angles before mixing 0° orientation plies. The 0° orientation plies may affect the contact stiffness, bending stiffness, damage extent, and compressive modulus. The compressive strength reaches the highest in tests on two samples with different percentages of 0° orientation plies and ply setups. Bending stiffness also dominates the impact response. The analysis on the laminate parameters provides ideas to improve the residual strength of helicoidal laminate.

## 1. Introduction

Composite materials such as fiber reinforced polymer (FRP) laminates are widely used in engineering structures with lightweight and high safety requirements, such as aerospace structures. Numerous composite structures are susceptible to impact from runway debris, hailstones, dropping tools, etc. The delamination caused by the low-velocity impact is generally invisible or barely visible; however, it reduces the flexural rigidity, thus leading to premature buckling failure under the in-plane compressive load [1,2]. Therefore, improving the damage resistance against impacts in FRP laminates is highly significant. Lamination sequences inspired by naturally occurring impact-resistant micro-structures, such as Bouligand structures [3] or helicoidal composite [4], have been investigated in the past years. A helicoidal composite with uniform inter-ply angles quasi-isotropic (QI) but some laminates are required to have directionally advantageous performance in stiffness or strength, such as the wing skin subjected to single directional bending. The helicoidal composites mixed with additional 0° plies were investigated in this study on its impact performance and residual strength under compression after impact (CAI).

Bouligand-featured structures can be found in a variety of animals, including mantis shrimp [5], fish scales [6], and the exoskeleton of arthropods, such as beetles [7]. The dactyl appendages of the mantis shrimps can endure forces of 0.4–1.5 kN, and the last segment of the appendage consists of several repetitions of the Bouligand unit stacked up along the thickness [5]. Each Bouligand unit of the dactyl appendage (Figure 1) contains a helicoidal layup with very small pitch angles (from 6.2° to 1.6°) [8,9] for a rotation of 180° inside each unit. Research in the field of biomechanics has shown that cracks growing in the matrix of the helicoidal composite specimens follow the fiber orientations and lead to the formation of twisted cracks [5]. A twisted crack growing in the helicoidal architecture amplifies the crack surface per unit volume, thereby enhancing energy dissipation and stress relaxation in the composite without leading to catastrophic failure. Some interlaminar cracks still exist; however, they are regarded as the sub-critical delamination rather than the large delamination occurring in ordinary composite.

Significant progress has been made in understanding the damage growth in Bouligand structures and studies have been conducted toward engineering materials. Suksangpanya et al. [10,11] developed a theoretical model for the local stress intensity factors at the crack front of twisting cracks formed within the Bouligand structure. Their study revealed that the changes in the local fracture mode at the crack front lead to a reduction of the local strain energy release rate, thereby increasing the energy required to propagate the crack, which is quantified by the local toughening factor. The theoretical values of the energy release rate and local stress intensity factors were validated using 3D simulations of the asymptotic crack front field. Mencattelli et al. [12] conducted a finite element (FE) simulation on thin-ply carbon fiber reinforced polymer (CFRP) laminates, thereby showing that the loci of maximum intralaminar shear stress infers the distribution of shearing matrix cracks in laminates with different pitch angles. Additionally, they investigated the effects of the pitch angle and ply thickness on the resistance of Bouligand CFRP laminates against the twisting cracks formed at the back face of the laminate subjected to low-velocity impact (LVI) due to the tensile stresses originating from bending.

In addition to the matrix cracks, penetration resistance is another key characteristic of composite panels. Using drop-weight impact tests, Grunenfelder et al. [8] demonstrated that the helicoidal composite panels exhibit higher penetration resistance with a wider-spread energy dissipation compared with the unidirectional and [0/±45/90] composite panels. The sinusoidally architected helicoidal laminate also plays a role in energy dissipation and increasing the load-bearing capability [13]. Another study conducted by Mencattelli et al. [14] reported that the laminates with a small pitch angle (2.5°, 5°, and 10°) have an overwhelming advantage over those with a pitch angle of 45° in terms of energy dissipation, peak load, and penetration load under quasi-static-indentation test. Liu et al. [15] reported that the resistance to the transverse load of helicoidal composite is related to the ratio of inter-ply angle to ply thickness and the strain energy release rate; they verified this using helicoidal laminates fabricated from unidirectional carbon-epoxy, Kevlar-epoxy, and glass fiber-epoxy prepregs [16]. The effect of pitch angle on the penetration energy of 3D printed helicoidal composite was studied by Liu et al. [17]; they found that the pitch angle of 5° is optimum.

According to the three-point-bending test, the resistance to the growth of cracks at the back face of the helicoidal laminate is less when the twist angle is small [14], so the inter-ply angle on the back face should be larger. Liu et al. [18] studied the effect of the non-uniform inter-ply angles on the penetration resistance and reported that the helicoidal configuration with a larger pitch angle (10°) at the bottom half and smaller pitch angle (5°) at the top half of the laminate consistently outperforms the other helicoidal laminates. Jiang et al. [19] conducted a numerical study on the helicoidal composite laminates with non-linear rotation angle; their study indicated that the opposite movement of adjacent layers may be hindered if they have different orientations.

In addition to the penetration resistance, residual strength under CAI is an important property of the laminates used in damage tolerance structures. Grunenfelder et al. [8] conducted CAI tests on the composite laminate; herein, the specimens were held in place by upper and lower clamp plates, with 76 mm diameter circular holes at the center, and an impact energy of 100 J was adopted. The compressive residual strength of specimen with a 7.8° pitch angle was 13 % lower than that of the QI specimen, whereas it was 16 % and 18 % higher than that of QI specimens with 16.3° and 25.7° pitch angles, respectively. Pinto et al. [20] reported that the symmetric helicoidal laminate has a better CAI strength compared with QI laminate; however, the pure helicoidal stacking configuration (in which the orientation of the plies goes from 0° to 360°) does not. In the study conducted by Ginzburg et al. [21], the residual compressive strength of CFRP helicoidal composite after high energy (80 J) impact was found to be better than that of the corresponding quasi-isotropic laminate; however, the residual compressive strengths of CFRP helicoidal laminates at lower impact energies (40 J) were lower than those of QI and cross-ply (CP) laminates. Mencattelli et al. [12] analyzed the compressive strength of samples with different pitch angles; however, no statistically significant trends were found.

Furthermore, studies on CAI of helicoidal laminates have also been conducted, and some studies demonstrate improvement in damage tolerance against foreign object impact by optimizing the mismatch angle between plies [22,23], bending stiffness [24], and thickness of ply [25,26,27] and relationship between pitch angle and thickness [28]. The CAI strength increases with the proportion of 0° plies (i.e., plies aligned with the loading direction) to some extent [29] for ordinary laminates. Sasikumar et al. [30] mixed thick plies of 0° with thin plies to fabricate a hybrid laminate, thereby yielding a 40% increase in CAI strength over the thin-ply baseline laminate. However, the proportion of 0° plies is constant if the helicoidal structure has a uniform pitch angle; hence, more 0° plies should be mixed into the laminate to increase its stiffness and strength in 0° orientation.

The compressive strength of laminates is generally tested according to ASTM D7136/D7136M-15 [31] and ASTM D7137/D7137M-17 [32], which were followed in this study. Impacts causing damage in samples are defined as low-velocity impacts when the contact time between the impactor and plate is sufficiently large to allow all wave reflections from the boundaries [33,34]; therefore, the LVI test in this study can be treated as a quasi-static problem. The performance of helicoidal composite under LVI could be describe with lumped-parameters model or energy-balance model [35,36]. In the model, the mass is mainly that of the impactor, and the stiffness is the contact stiffness between the impactor and laminate, the bending stiffness, and the membrane stiffness of the laminate. The contact stiffness is relative to the modulus and local strength under the concentrated load [37,38,39]. The bending stiffness is relative to the bending stiffness matrix coefficients of the laminates [40,41,42,43]. These material parameters relative to the ply setup need to be considered to study the properties of the composite laminate to resist LVI or vibrational load [44,45].

In this study, helicoidal composite materials were mixed with additional 0° plies in order to increase the CAI strength, and the pitch angles of some plies were designed to make the laminates symmetric for the fixed ply number. Subsequently, helicoidal laminates with non-linear twist angle (NLTA) were yielded. Additionally, helicoidal laminates with constant pitch angle, QI laminates, and CP laminates were fabricated for a comparative investigation. The damage modes (after LVI and CAI tests) and residual strength were analyzed based on impact theory, shear stress simulation, and strain energy release rate.

## 2. Materials and Experimental Setup

### 2.1. Specimens Design

Helicoidal, QI, and CP laminates were designed and fabricated using unidirectional T700/SKRK51 carbon-epoxy prepreg, the mechanical properties of which are shown in Table 1. The average ply thickness is 68.5 μm. The total number of layers used in this study was fixed at 73 for each plate leading to an overall plate thickness of 5 mm. The minimum pitch angle
Δθ was set to 2.5°. The configurations and abbreviation used to denote them are provided in Table 2, where H denotes the helicoidal laminate, QI denotes the quasi-isotropic laminate, and CP denotes the cross-ply laminate. We manufactured 2 samples for each type of laminate. All of the layups were symmetric to avoid the thermally induced twist in the laminate in the case of the non-symmetric layup. Layups H1, H2, and H3 consisted of a uniform pitch angle; therefore, the plies had a linear twist angle (LTA). The other helicoidal layups were mixed with more 0° plies in addition to the origin 0° plies of the LTA laminate to maintain the continuity. The ply orientation of H4, H5, and H6 are demonstrated in Figure 2.

Herein, H4 had a basic pitch angle of 5°. Five consecutive 0° plies were on two surfaces, and one was in the middle plane (180°). To achieve a symmetric sequence, a pitch angle of 10° was applied to plies 20–23 and 52–55. Layup H5 had a basic pitch angle of 10° and 12 plies of 0° (180°). A pitch angle of 20° was applied to plies 11, 12, 29, 45, 46, 63, and 64 to achieve a symmetric sequence. Layup H6 had pitch angles of 2.5°, 5°, and 10°. Eleven plies were of 0° (180°), and the pitch angle was 2.5° for the plies close to the 0° (180°) plies, which increased the stiffness in that direction while maintaining a non-zero pitch angle.

The extensional stiffness coefficients and bending stiffness coefficients were calculated, see Table 2. Specimen H1, H2, and H3 are QI laminates according to the extensional stiffness coefficients. More 0° plies increase $A_{11}$ for specimen H4, H5, and H6, while CP-73 has the maximum $A_{11}$ due to possessing the most 0° plies.

The plies laid in the required configurations were cured using an autoclave at 125 °C and 5 bars for 90 min. The cured laminations were cut into 150 mm × 100 mm squares to meet the testing standard ASTM D7136.

### 2.2. Test Equipment

Low-velocity-impact (LVI) tests were conducted using Instron CEAST 9350 drop-weight tower (Figure 3a) following ASTM D7136. The impactor mass was 5.277 kg and the hemispherical tip was 12.7 mm in diameter. To achieve a reasonable amount of damage, each sample was impacted with an impact energy of 20 J, i.e., 4 J/mm along the thickness. The time history of impact load, displacement of the impactor, and energy absorbed by the specimen were recorded by the control system of the tester. After the LVI test, the dent depth was measured using a vernier micrometer. The computed tomography (CT) images were obtained from Zeiss Xradia 520 Versa X-ray microscope (Zeiss, Oberkochen/Jena, Germany) (Figure 3c).

A set of compression after impact (CAI) tests was performed on the samples following the testing standard ASTM D7137 (Figure 3b). Samples were loaded by a 300 kN load cell tester with a constant displacement rate of 1.25 mm/min. The strains during the test were measured by four strain gauges instrumented on the front and back sides of the samples to verify that the initial alignment of the sample was within 10% and that no global buckling occurred. The ultimate compressive residual strength and effective compressive modulus were recorded in the test direction.

## 3. Results and Discussion

### 3.1. Experimental Results

The outputs of the operating system were the time history of the impact load, curve of impact load versus the displacement of the impactor, and time history of energy absorbed by the specimen (see Figure 4), which are used to analyze the characteristics of different specimens in Section 4. The images of the impact site and distal face are shown in Figure 5, some of which reflect the internal damage. The ultrasonic C-scanning images were obtained using GE Phasor XS, as shown in Figure 6, where the blue area denotes the delamination region to be discussed in Section 3.3 and Section 3.4. The side cross-sectional views of CT images are shown in Figure 7.

The peak load, maximum displacement of the impactor, final energy absorbed due to sample damage, and the indentation measured after the impact test are shown in Figure 8a–d, respectively. Furthermore, the bending stiffness of each specimen was calculated to reveal the characteristics of the laminate under LVI. The CAI strength was plotted against the bending stiffness and compressive modulus, as shown in Figure 9a,b, respectively.

According to the experimental results, the impact load, damage area, strain energy release rate, and CAI characteristics are analyzed in the following subsections, respectively.

### 3.2. Impact Load

According to the lumped-parameter model [36], the impact load is affected by stiffness. The stiffness is relative to the bending stiffness, membrane stiffness, and contact stiffness, and the bending stiffness is dominated by the stack sequence. The spring stiffness in the lumped-parameter model related to the laminate bending stiffness (i.e., the main part of transverse stiffness when the plate is normally applied a concentrated load) is denoted by $K_{b}$ (in N/m) and may be expressed from Navier’s solution [43]:(1)$$K_{b}=\frac{\bar{A}\pi^{4}}{4b^{2}}\frac{1}{\sum_{m=1}^{\infty }\sum_{n=1}^{\infty }\frac{1}{\left[D_{11}{\left(\frac{m}{\bar{A}}\right)}^{4}+\frac{2\left(D_{12}+2D_{66}\right)m^{2}n^{2}}{{\bar{A}}^{2}}+D_{22}n^{4}\right]}}$$
where $\bar{A}=a/b$, a and b are the length and width of the rectangle boundary of the laminate sample, and are equal to 125 mm and 75 mm, respectively, $D_{ij}$ represents the elements of the bending stiffness matrix; and m and n are odd. Equation (1) implies that the coefficients $D_{16}$ and $D_{26}$ are zero; however, they are not equal to zero for some helicoidal samples used in this study. Though Equation (1) is not accurate for all samples, it was employed to evaluate the effect of the ply setup on the impact response. For each laminate, $K_{b}$ was calculated and plotted (Figure 8 and Figure 9).

The transverse stiffness affects the response when the laminate is subjected to a concentrated load. Figure 8a shows that the trend of peak load follows that of $K_{b}$, and it is similar to that of the energy absorption due to damage (Figure 8c) as a higher load leads to serious damage. A higher peak load coincides with a lower impactor displacement when the impact energy is the same; therefore, the trend of impactor displacement is opposite to that of $K_{b}$ (Figure 8b). It can be seen from curves shown in Figure 4b that H4 and H6 have a lower peak load and a larger displacement, which also have the lower $K_{b}$.

The indentation is affected primarily by transverse modulus and matrix cracks. It decreases with an increase in the pitch angle of the LTA laminates: H1, H2, and H3, as shown in Figure 8d, because the small angle plies are easy to be pushed away by the hemispheric impactor tip [19]. Consequently, H4 and H5 also have a high indentation, because of five 0° plies arranged on the surfaces. The specimen CP-73 has a higher indentation, which may be caused by local damage. A higher indentation decreases the contact load, thus leading to the opposite trend of the two parameters for each specimen, as shown in Figure 8a,d. Figure 8d indicates that the trend of indentation is also opposite to that of $K_{b}$. Even though a high stiffness causes a higher peak load and a deeper indentation, the indentation is more dependent on the ply setup near the surface; hence, a small pitch angle leads to deeper indentation.

The trend of CAI strength is irrelative to that of bending stiffness (Figure 9a) and compressive modulus (Figure 9b), which is a result of the complicated damage initiation and its propagation.

The transverse stiffness which allows laminate to resist the impact load, and to which the peak load is mainly relative, was studied. Transverse stiffness is defined as the ratio of the impact load and the displacement of the impactor measured in the test; it is also a result of bending stiffness, membrane stiffness, and contact stiffness. During impact, delamination and other damages occur, thus leading to a decrease in transverse stiffness. Therefore, two values of transverse stiffness corresponding to before and after delaminating, namely $K_{0}$ and $K_{d},$ respectively, are introduced to demonstrate the behaviors of the laminate. The load curve versus the displacement of the impactor on H1, having two segments of different slopes in the loading procedure, is shown in Figure 10. The slopes were determined by linear fitting. The stiffness of each laminate and the residual stiffness in percentage is shown in Figure 11a. A comparison between transverse stiffness and bending stiffness is shown in Figure 11b.

The transverse stiffness $K_{0}$ increases with an increase in twist angle of LTA specimens: H1, H2, and H3 (Figure 11a); however, the residual stiffness, $K_{d}$, does not. Specimens H4 and H6 have a higher percentage of residual stiffness, because the initial stiffness is low and more impact energy is absorbed by elastic deflection rather than by damage growth (Figure 8c). The LTA laminates, namely H1, H2, H3, QI-73, and CP-73, possess higher values of $K_{0}$. Laminate H5 also has a high $K_{0}$ because the plies in 0° direction were set on a depth of 2/7 and 5/7 beside the two surfaces, and the contact stiffness is high when a small indentation is observed (Figure 8d). These factors make H5 possess a higher transverse stiffness than the other two NLTA specimens. The higher transverse stiffness of H5 allows it to absorb more energy compared with H4 and H6 upon damage growth (Figure 8c). Additionally, the percentage of residual stiffness in H5 is lower, which indicates serious damage (Figure 11a); $K_{0}$ goes with $K_{b}$ roughly, however $K_{d}$ does not (Figure 11b).

### 3.3. Damage Area

The helicoidal damage is evident from the ultrasonic C-scan images of H1 (Figure 6a; the figure reveals two sectors of damage, which are the projections of two cycles of twisting Bouligand-like cracks shown by the CT image (Figure 7a)). Similarly, four and eight cycles of twisting Bouligand-like cracks are presented in H2 and H3, respectively; hence, the boundary of their C-scan images is smooth.

The C-scan images of NLTA specimen, H4, and H6 indicate that the damage is extended along the 0° direction owing to multiple 0° layers on the damage-prone upper and lower surfaces. The 3D reconstruction of the CT image of H4, shown in Figure 12a, shows multiple cracks in the 0° direction on the back. Additionally, a large crack on the distal face is observed in the images shown in Figure 5d. These cracks expand along the 0° layer, and then develop in a spiral shape. The cracks in the 0° direction form a relatively large projection area in the C-scan image, so the number of continues 0° plies on the impact surface should be limited.

The specimen with the smallest pitch angle, H1, has no such long crack on the back of the impact point. A crack in the 90° orientation appears near the impact point on the surface, see Figure 5a. The CT scan image, shown in Figure 12b, indicates that there is only one crack on the impact surface that develops into a spiral crack after expanding to a certain depth.

The damage areas from C-scanning of QI-73 and CP-73 are small; however, Figure 8c indicates that as these two test pieces absorbed more energy due to damage, the damage is widely distributed. The CT scan section image near the impact point shown in Figure 7 shows that the delamination of QI-73 and CP-73 is very dense. The delamination of spiral laminates is related to the number of cycles and H1 has only two cycles. However, the crack could easily propagate to the adjacent matrix owing to the small pitch angle. Specimen H2 has four cycles of delamination, whereas H3 has eight cycles. The non-linear layups, H4 and H6, have delamination similar to that of H1 and also have scattered cracks. The delamination position of specimen H5 is similar to that of H2.

### 3.4. Analysis on Damage Shape with Strain Energy Release Rate and Shear Stress

The role of pitch angle in the growth of twisting Bouligand-like cracks was analyzed using numerical methods. Suksangpanya et al. [10,11] developed a model for local stress intensity factors along the crack front of a continuously twisting crack in a Bouligand medium, originating from an initially flat crack from the distal face due to the tensile stresses developed from bending (Figure 5 in [10]). Mencattelli et al. [12] computed the strain energy release rate for the growth of twisting Bouligand-like cracks using this model and demonstrated the relationship between the damage feature and pitch angle and the ply thickness. The distribution of maximum $\tau_{23}$ stress calculated using the FE model shows the damage distribution in the Bouligand CFRP thin-ply laminates. In this section, the FE results are also presented.

According to Figure 5 in [10], the twisting of the crack front can be represented by the angle (ϕ) between Z and $z^{\prime }$, and for a fiber-reinforced laminate with an individual ply thickness of $t_{\mathrm{ply}}$, it is given by:(2)$$\phi =X\left(\frac{\Delta \theta }{t_{\mathrm{ply}}}\right)$$

In addition, kinking of the crack is represented by the effective kink angle $\alpha^{*}$, which is defined as the angle between $y^{*}$ and *Y*, where $y^{*}$ is obtained by rotating y about *X* by an angle—ϕ, and can be expressed as:(3)$$\alpha^{*}=\cos^{-1}\left\{\frac{\cos\phi \left[1+\tan^{2}\phi \right]}{\sqrt{{\left[(Z\Delta \theta /t_{\mathrm{ply}})\sec^{2}\phi \right]}^{2}+1+\tan^{2}\phi }}\right\}$$

In the context of four assumptions taken in their study, Suksangpanya et al. [10] provided the energy release rate $\left(G\right)$ at each point on the twisting crack front, and normalized to the strain energy release rate ($G_{0}$) in the case of the crack remaining flat, as follows:(4)$$\frac{G}{G_{0}}={\left(\frac{k_{I}^{\prime }}{K_{I}}\right)}^{2}+{\left(\frac{k_{II}^{\prime }}{K_{I}}\right)}^{2}+\frac{1}{1-v}{\left(\frac{k_{III}^{\prime }}{K_{I}}\right)}^{2}$$
where v is the Poisson ratio of the isotropic medium, and the local stress intensity factors along the crack front $k_{I}$, $k_{\mathrm{II}}$, and $k_{\mathrm{III}}$ are expressed as:(5)$$\begin{array}{l} \frac{k_{I}^{\prime }}{K_{I}}=\cos\left(\frac{\alpha^{*}}{2}\right)\left[\cos^{2}\left(\frac{\alpha^{*}}{2}\right)\cos^{2}\phi +2v\sin^{2}\phi \right]\\ \frac{k_{II}^{\prime }}{K_{I}}=\sin\left(\frac{\alpha^{*}}{2}\right)\cos^{2}\left(\frac{\alpha^{*}}{2}\right)\cos\phi \\ \frac{k_{III}^{\prime }}{K_{I}}=\cos\left(\frac{\alpha^{*}}{2}\right)\sin\phi \cos\phi \left[\cos^{2}\left(\frac{\alpha^{*}}{2}\right)-2v\right] \end{array}$$

Assuming Z = 0.5 mm, the normalized energy release rate from the 1st to 37th ply of every sample was calculated (Figure 13). Evidently, the energy release rate decreased with an increase in pitch angle H1, H2, and H3. A higher pitch angle tends to resist the propagation of intra-lamina cracks, thus leading to more catastrophic mechanisms of failure such as delamination and fiber breaks. Samples H4, H5, and H6 possessed several consecutive 0° plies, thus leading to a higher energy release rate at the relative layers of consecutive 0° plies. Except at consecutive 0° plies, the energy release rates of H4 and H5 coincided with that of H1 and H2, respectively. H6 had a pitch angle of 2.5° for some plies, thus leading to a higher energy release rate than others at most positions, followed by H4, with a pitch angle of 5°. The energy absorption due to damage in H4 and H6 is lower than that of other specimens (Figure 8c) owing to their higher energy dissipation due to scatter cracks and lower transverse stiffness.

The distribution of the maximum intra-ply shear stress ($\left|\tau_{23}\right|$) within the test samples was used to demonstrate the delamination development following the method outlined by Mencattelli et al. [12]. Six static linear elastic models were developed for each helicoidal composite specimen using the FE software Abaqus 2020. Geometries of both the laminate and indenter used were the same as the ones used in actual testing conditions. The material properties assigned to each lamina are reported in Table 1. Solid elements (C3D8R) were used to model the laminate (ply-by-ply using lamination theory) and a rigid surface for the indenter (Figure 14), with a minimum size of 1.4 mm. The mesh was refined in the central region of the laminate, with a minimum size of 0.2 mm after mesh convergence analysis. Analysis showed that large mesh sizes, such as 0.4 mm, result in location discrepancies in the maximum shear stress at each layer, especially if the deviation angle is small. Integration points for adjacent layers may overlap, resulting in approximations and reduced accuracy, with a maximum stress point to a distance error of up to 19%. A frictionless hard contact was created between the upper surface of the composite laminate and the outer surface of the indenter.

Post-processing was performed using a Python script to extract the maximum shear stress and corresponding the integration point locations from each ply of the laminate. The locations of the maximum $\left|\tau_{23}\right|$ are plotted in 3D mode and also projected to the plane of the laminate (Figure 15), where 0° ply is aligned with the x-axis. For each ply, Figure 15 shows the likely location of the shearing cracks, i.e., the points with the maximum $\left|\tau_{23}\right|$. In each ply, $\left|\tau_{23}\right|$ exhibits two identical maximum values. The loci of the points with maximum $\left|\tau_{23}\right|$ form a double helix in H1, H2, and H3 (Figure 15a–c). The double helices are symmetric about the laminate symmetry plane. Some drastic changes in the locations of the maximum $\left|\tau_{23}\right|$ in H4 and H6 are due to the consecutive 0° plies. When observed perpendicularly to the plane of the laminate, the out profile of the loci of the maximum $\left|\tau_{23}\right|$ reflects the shape of C-scan image shown in Figure 6. The sector delaminated region is obvious for H1 and H2, and the protruding part on C-scan image of H4 and H6 due to the multiple 0° ply is clearly shown by $\left|\tau_{23}\right|$.

### 3.5. CAI Characteristics

Figure 9a indicates that the relationship between CAI strength and $K_{d}$ is uncertain. The laminates with high compressive modulus possess high CAI strength except for H1 and H4. The pitch angle of H1 is 5°, whereas H4 has two pitch angles of 5° and 10°. The constraints between the plies with small pitch angles are weak. The indentation of H1 is also greater than that of H2 and H3 due to the weak interlaminar constraints, which means the more serious local damage or fiber distortion; hence, it is more susceptible to damage during compression. This could be regarded as a characteristic of small pitch-angled composites. CAI strength is less affected by the extent of impact damage here, as shown in Figure 8c. The energy absorbed due to damage is nearly equal for H1, H2, and H3; however, their CAI strengths (and that of H4 and H6) are different.

To further investigate the characteristics of specimens, the critical buckling load, $N_{x,\mathrm{cr}}$, applied on the 150 mm × 100 mm specimen with four edges, simply supported along the longer edge, was computed using the FE model, as listed in the last row of Table 2. Note that the value of the critical load is meaningful only numerically because the stresses corresponding to the buckling loads have exceeded the strength of the material in the stability analysis of plates of such sizes. H2 and H3 have a higher buckling load than H1 and therefore have the higher CAI strength under the same damage extent.

Figure 9b shows that the compressive modulus of H4 is higher than that of H2 and H3 because H4 has more 0° plies; however, its CAI strength is lower. Samples H1, H2, H3, and QI-73 are quasi-isotropic laminates; thus, the compressive modulus is similar. However, the CAI strengths of H2 and H3 are higher than that of QI-73, thereby indicating the better performance of helicoidal composites with 10° and 20° pitch angles.

The mean CAI strengths of H5 and H6 are higher than those of QI-73 and CP-73. Its value for H5 is slightly higher than that of H6; however, the dispersion of CAI strength in H6 is higher. H5 has pitch angles of 10° and 20°, whereas H6 has 2.5°, 5°, and 10°. H6 has the highest compressive modulus and lower transverse stiffness (Figure 11a) to resist the impactor. This results in a low peak load (Figure 8a) and more energy absorption by elastic deformation than by damage (Figure 8c), eventually leading to a high CAI strength.

The stiffness, $k_{b}$, which is relative to transverse stiffness, is calculated from the bending stiffness coefficients $D_{11}$ and $D_{22}$. Therefore, CAI strength and compressive modulus are plotted against $D_{11}$ and $D_{22}$ (Figure 9c,d). A definite relationship between CAI and bending stiffness coefficients is not obtained from Figure 9c. In Figure 9d, the compressive modulus roughly follows $D_{11}$, as they are strongly affected by the number of plies in the 0° orientation.

The transverse stiffness and peak load of H5 are higher than that of H6. The CAI strength and energy absorbed due to damage are also higher in H5 compared with H6; however, the compressive modulus of H5 is lower than that of H6. This is attributed to the proper pitch angles of 10° and 20°, and the enhancement of CAI strength by 0° orientation plies in H5 despite being less in number compared to H6. The critical buckling loads are 2100.8 N/mm and 1569.2 N/mm for sample H5 and H6, respectively; the results revealed that H5 is superior to H6 in terms of structural stability.

## 4. Conclusions

Helicoidal composite laminates made from carbon-epoxy prepreg with plies reinforcing the loading capability in 0° direction were studied using low-velocity impact tests and compressive tests. In this study, we can find that:The laminates having pitch angles of 10° and 20° were found to possess higher CAI strengths. However, having a large number of 0° plies yielded a larger damage area because of the easy propagation of cracks in them. The continuous 0° plies on the impact surface also yielded larger indentation, thereby leading to a decrease in the transverse stiffness and the impact load. Lower impact load resulted in less damage, which could be estimated from the energy absorption.The specimen with the plies of 10° and 20° pitch angles and continuous 0° plies (H5) yielded the highest CAI strength despite more damage than the one with more plies in 0° and a higher compressive modulus (H6), because a small pitch angle (5°) should be avoided due to the weak constraint between adjacent plies. Therefore, a set of proper pitch angles (10° and 20°) and proper arrangement of 0° plies are the key parameters for CAI strength.Analysis on the strain energy release rate and maximum $\left|\tau_{23}\right|$ could be beneficial in the prediction of damage characteristics.

The first ply in all specimens is of 0°; therefore, a method should be proposed to evaluate the effect of the first ply direction on its performance. Non-symmetric and non-uniform layup with low coupling stiffness could be optimized to obtain better performance.The mechanism of ply pitch angle affecting impact load, indentation, damage, CAI strength, and critical buckling load needs further investigation to obtain the optimal ply setups for a variety of services.

## Figures and Tables

**Figure 1 materials-16-04599-f001:**
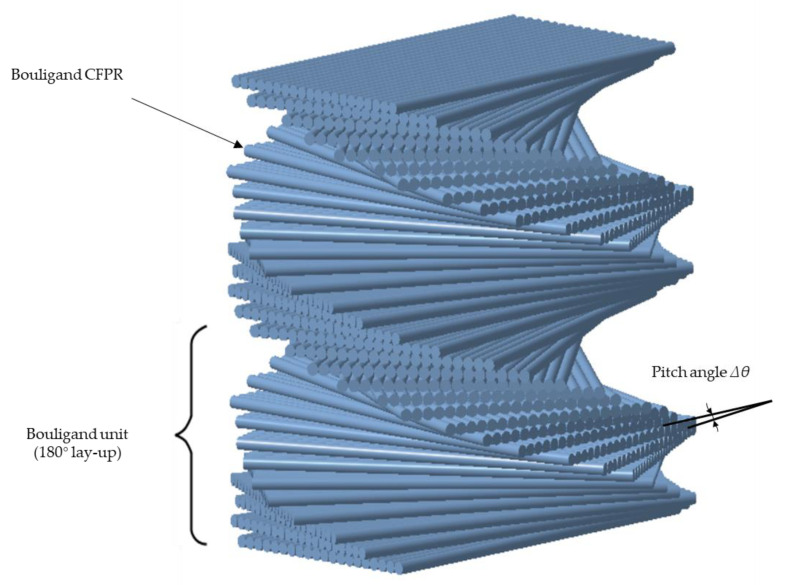
3D schematic of the Bouligand structure in the periodic region of the dactyl club.

**Figure 2 materials-16-04599-f002:**
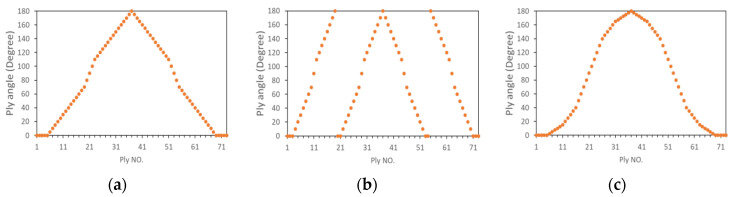
Orientation of the plies for samples (**a**) H4, (**b**) H5, and (**c**) H6.

**Figure 3 materials-16-04599-f003:**
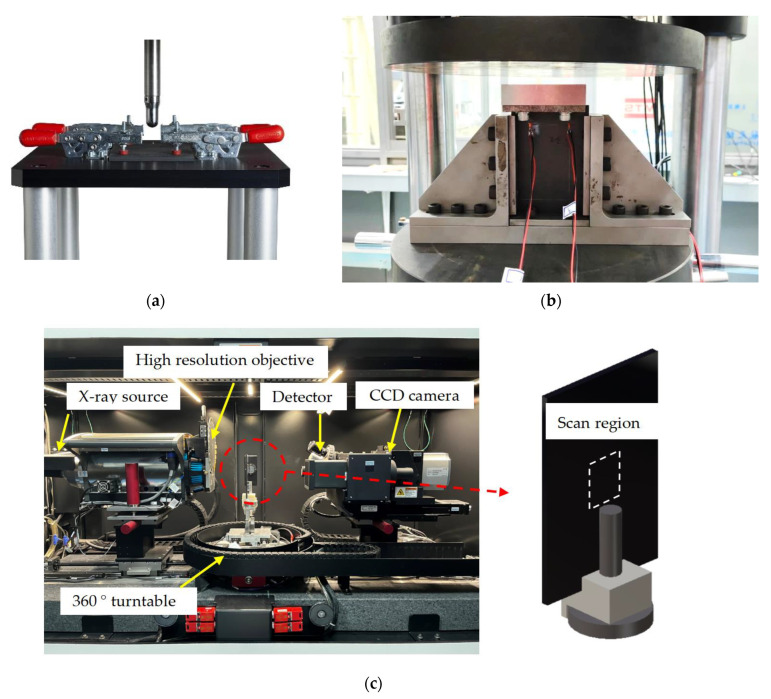
Experimental setup: (**a**) LVI test, (**b**) CAI test, and (**c**) Zeiss Xradia 520 Versa X-ray microscope for CT imaging and installation of specimen.

**Figure 4 materials-16-04599-f004:**
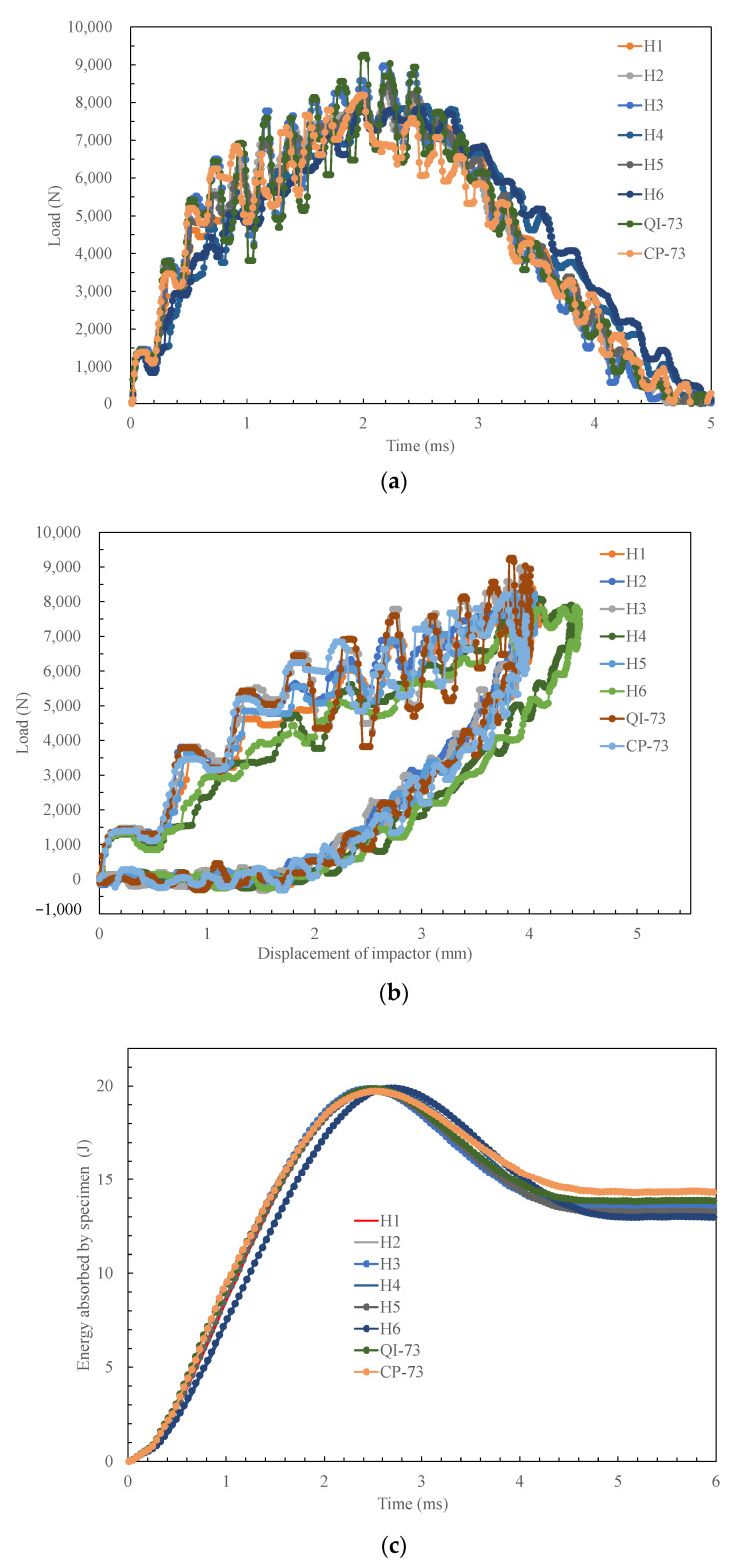
Experimental data of low-velocity test: (**a**) time histories of the impact load, (**b**) displacement of impactor vs. impact load, and (**c**) time histories of the energy absorbed by specimen.

**Figure 5 materials-16-04599-f005:**
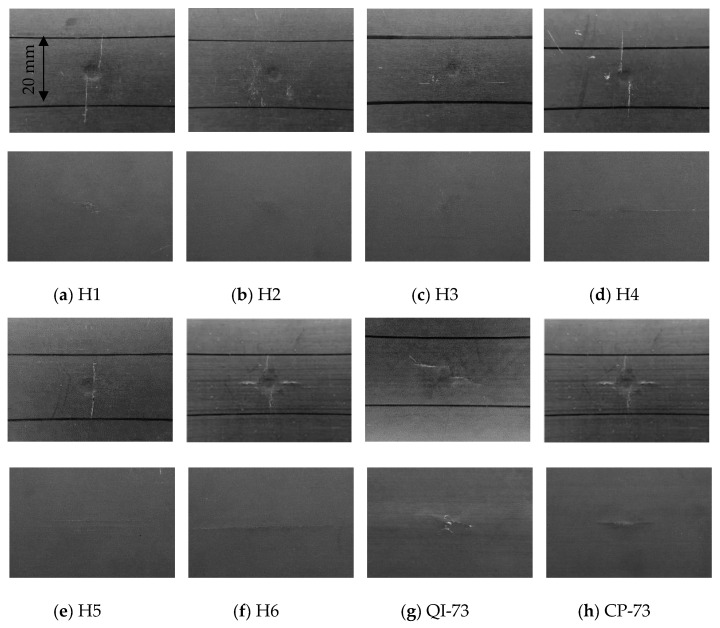
Images of the impact site (**upper**) and distal surface (**lower**) of specimens.

**Figure 6 materials-16-04599-f006:**
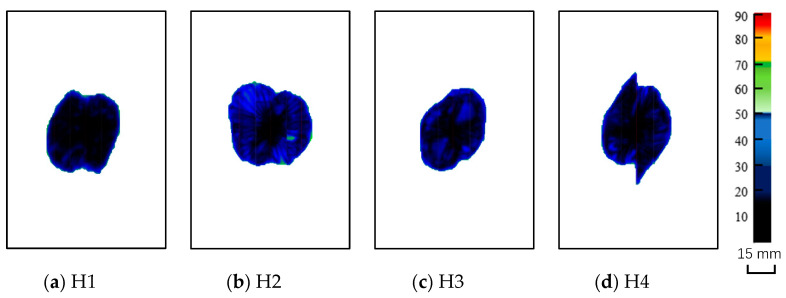
Ultrasonic C-scan images of the composite laminates.

**Figure 7 materials-16-04599-f007:**
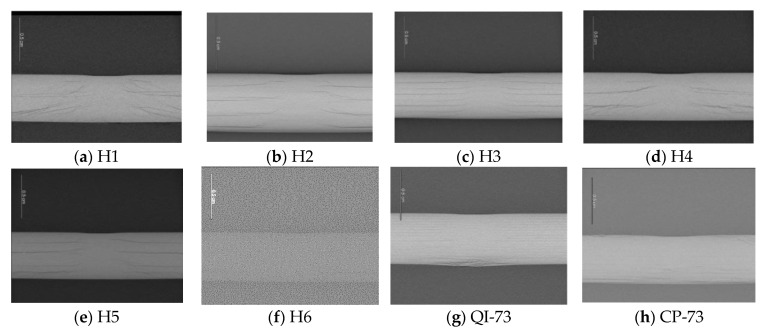
Side cross-sectional view of the central area of laminates.

**Figure 8 materials-16-04599-f008:**
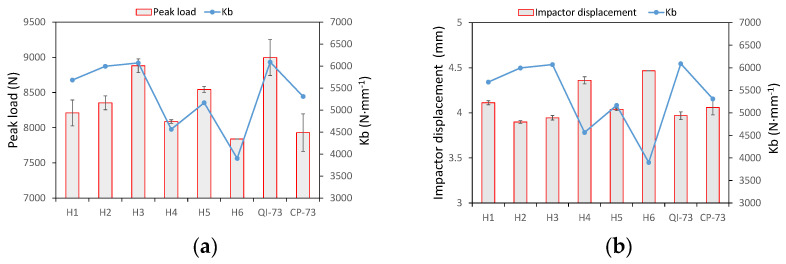
Relationships between (**a**) peak load, (**b**) impactor displacement, (**c**) energy absorbed, and (**d**) indentation of impact tests and $K_{b}$ of samples.

**Figure 9 materials-16-04599-f009:**
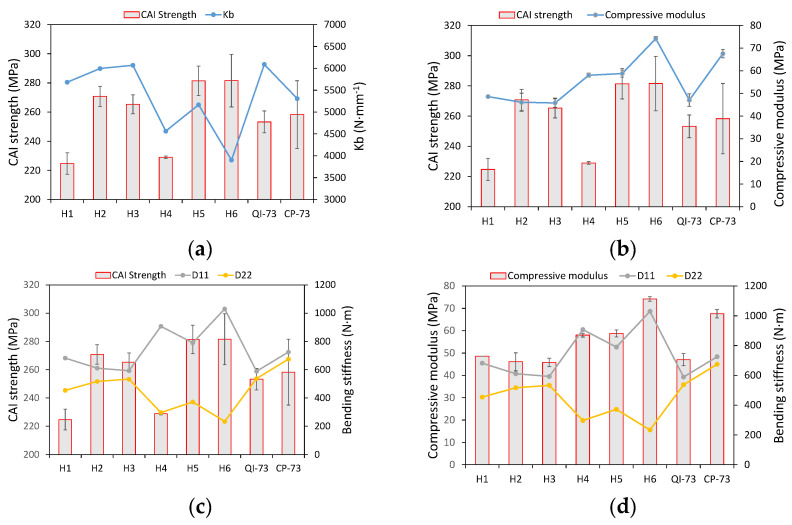
Relationship between CAI strength and (**a**) $K_{b}$, (**b**) compressive modulus, (**c**) bending stiffness, and (**d**) relationship between compressive modulus and bending stiffness.

**Figure 10 materials-16-04599-f010:**
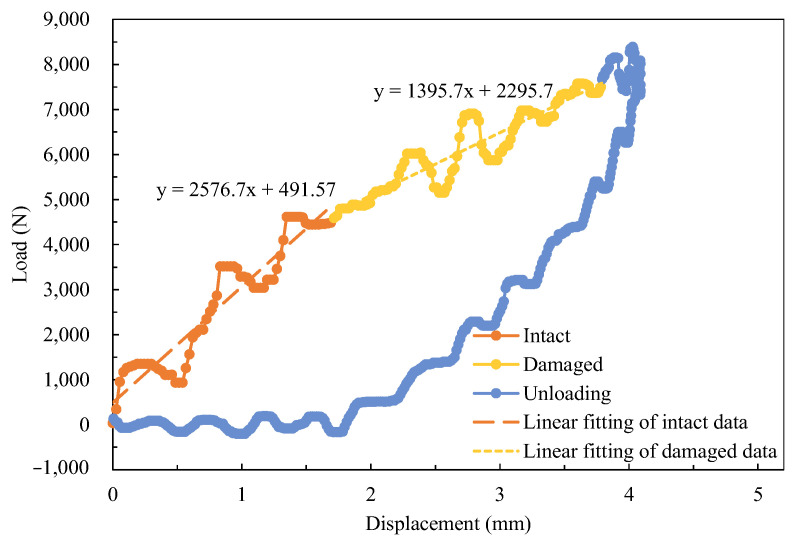
Load–displacement curve for the determination of transverse stiffness for sample H1.

**Figure 11 materials-16-04599-f011:**
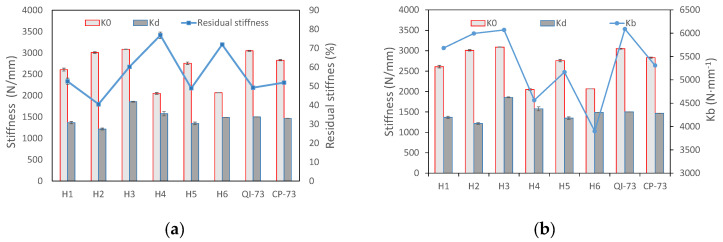
Tendency of transverse stiffness with (**a**) residual stiffness in percentage and (**b**) $K_{b}$.

**Figure 12 materials-16-04599-f012:**
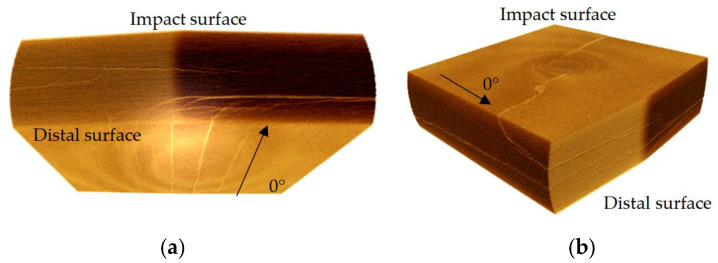
Cracks on (**a**) the distal face of H4 and (**b**) on the impact face of H1.

**Figure 13 materials-16-04599-f013:**
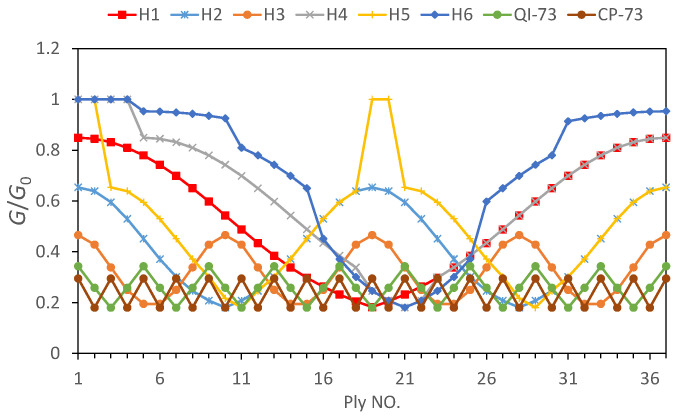
Energy release rate ($G/G_{0}$) vs. X evaluated at Z = 0.5 mm for each ply of the specimen with a different ply sequence.

**Figure 14 materials-16-04599-f014:**
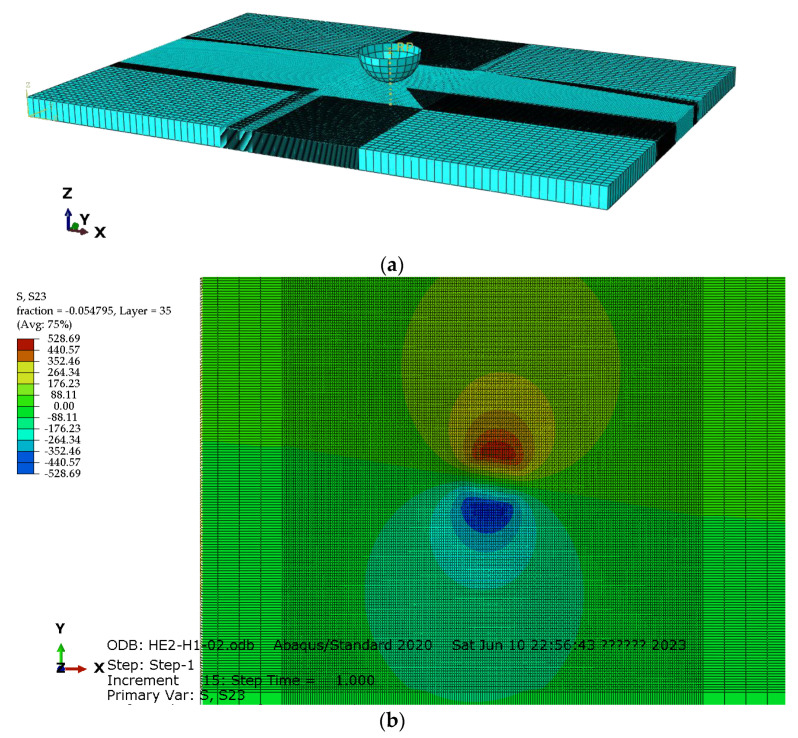
FE model of the specimen for the simulation of quasi-static indentation test: (**a**) global view to the model and (**b**) shear stress distribution of a ply in H1.

**Figure 15 materials-16-04599-f015:**
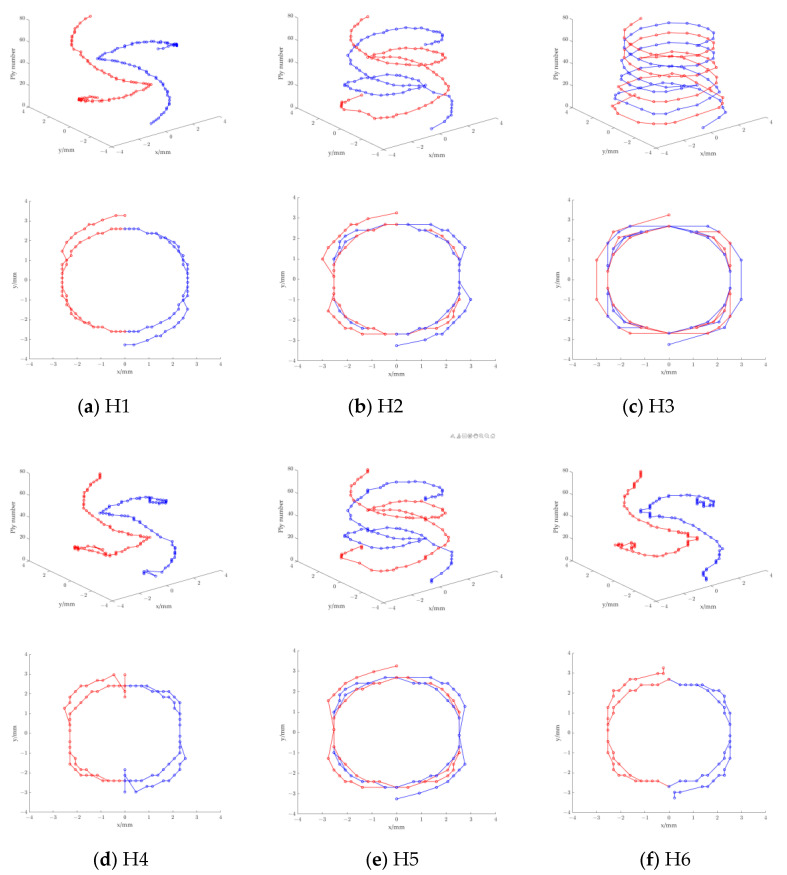
Loci of maximum $\left|\tau_{23}\right|$ (**upper**) and its projection on the plane of the laminates (**lower**).

**Table 1 materials-16-04599-t001:** Mechanical properties of T700/SKRK51 carbon-epoxy prepreg.

Mechanical Properties	Value
0°Tensile modulus $E_{1}$ (GPa)	125.3
90°Tensile modulus $E_{2}$ (GPa)	8.4
0°Tensile strength (MPa)	2500
0°Compressive strength (MPa)	780
90°Tensile strength (MPa)	60
0°Interlaminar shear strength (MPa)	88.5

**Table 2 materials-16-04599-t002:** Sample specifications.

Samples	Stacking Sequence	Extensional Stiffness Coefficients (MPa)			Coefficients of Bending Stiffness Matrix (N·m)						The Critical Load of Buckling (N/mm)
		$A_{11}$	$A_{12}$	$A_{66}$	$D_{11}$	$D_{12}$	$D_{22}$	$D_{66}$	$D_{16}$	$D_{26}$	$N_{x,cr}$
H1	[0/5/10/…/170/175/$\bar{180}$]s	272	264	92	682	160	454	182	172	107	2115
H2	[0/10/20/…/160/170/180/10/20/…/170/$\bar{180}$]s	272	264	92	611	164	517	186	85	54	2348
H3	[0/20/40/…/160/180/20/40/…/160/180/20/…/160/180/20/…/160/$\bar{180}$]s	272	264	92	593	165	533	187	40	27	2385
H4	[0/0/0/0/0/5/10/15/20/…/60/65/70/80/90/100/110/115/120/…/165/170/175/$\bar{180}$]s	304	233	91	907	126	297	148	125	68	1771
H5	[0/0/0/10/20/30/…/60/70/90/110/120/…/150/160/170/180/0/0/10/20/30/…/60/70/90/110/120/…/150/160/170/$\bar{180}$]s	336	204	90	790	147	372	169	66	36	2101
H6	[0/0/0/0/0/2.5/5/7.5/10/12.5/15/20/25/30/35/40/50/60/…/120/130/140/145/150/155/160/165/167.5/170/172.5/175/177.5/$\bar{180}$]s	415	159	73	1030	96	234	118	104	35	1569
QI-73	[(0/45/90/−45)_9_/0/(−45/90/45/0)_9_]	272	264	92	589	165	537	187	12	12	1901
CP-73	[(0/90)_36_/0]	340	332	24	725	28	675	50	0	0	2394

## Data Availability

Not applicable.

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
