# Peer review of "Increasing the Compressive Strength of Helicoidal Laminates after Low-Velocity Impact upon Mixing with 0° Orientation Plies and Its Analysis"

_materials, 2023, doi:10.3390/ma16134599_

Round 1

Reviewer 1 Report

Please see the comments below.

1.       Please label important components of Fig. 4(a) for better understanding

2.       If available, please add color-scale for the interpretation of color in ultrasonic C-scan images (Fig. 9).

3.       Section 4.1 heading (line 234), “impact” is misspelled as “Iimpact”. Please correct.

4.    Please further elaborate why the pitch angles especially 10 and 20 degree lead to higher CAI strength.

Author Response

  1. Please label important components of Fig. 4(a) for better understanding

Reply: we labeled main components.

  1. If available, please add color-scale for the interpretation of color in ultrasonic C-scan images (Fig. 9).

Reply: We added the color-scale. We obtain the delamination using GE Phasor XS scanner, which is depicted in the revised version. Only the location with the reflection energy lower than a threshold was recorded. So damage depth was not reflected accurately.

  1. Section 4.1 heading (line 234), “impact” is misspelled as “Iimpact”. Please correct.

Reply: we corrected it, thanks.

  1. Please further elaborate why the pitch angles especially 10 and 20 degree lead to higher CAI strength.

Reply:

1)In section 4.4, we added:

The indentation of H1 is also greater than that of H2 and H3 due to the weak interlaminar constraints, which means the more serious local damage or fiber distortion

2)We calculated all of the buckling load in table2, and conducted an analysis in Section 4.4:

To furtherly investigate the characteristics of specimens, the critical buckling load, , applied on the 150 mm×100 mm specimen with four edges, simply supported along the longer edge, was computed using the FE model, as listed in the last row of Table 2. H2 and H3 have a higher buckling load than H1, therefore have the higher CAI strength under the same damage extent.

Reviewer 2 Report

The paper regards the investigation of the compressive behavior of laminates after impact, with emphasis on the damage produced by the impact.

The revised paper should improve some aspects of the method description and results presentation:

1.       Section 2 sounds like a laboratory report, to better fit the purpose of a scientific article the purposes of the choices made for the experimental tests should be discussed.

2.       The results reported in Figure 6 to Figure 9 need to be commented on and discussed.

3.       Is it possible to justify the evidence discussed on page 11, line 284 about the different trends of the stiffness K_0 and K_d with the twist angle?

4.       To give an idea about the accuracy of the stiffness value given by equation (1), the percentage difference K_b - K_0 could be added to Figure 14.

5.       Which data from the FE analysis was used to obtain the results in Figure 18?

6.       Regarding the FE model, some aspects need to be clarified

a.       Give details about the contact pairs between the plate and the indenter.

b.       How did you evaluate the mesh convergence of the model?

c.       Considering that the simulation regards a contact problem, the analysis should not be quasi-static but explicit in order to take into account the dynamic effects that in this case cannot be neglected.

d.       The Von Mises stress is not adequate to measure the equivalent stress state in composite laminates. Another contour plot should be reported.

7.       Regarding the FE buckling analysis in section 4.4, how did you consider the damage related to the impact? Were the elastic properties of the plates reduced? If the damage was not considered these analyses cannot be correlated with compression after impact tests.

8.       Carefully check the paper to correct typos and misspells.

9.       The references should be enlarged with more papers regarding the bending stiffness of laminates, such as: https://doi.org/10.1016/j.compstruct.2013.12.011; https://doi.org/10.1016/j.tws.2022.110374; https://doi.org/10.1016/j.apm.2022.06.012

 Carefully check the paper to correct typos and misspells.

Author Response

  1. Section 2 sounds like a laboratory report, to better fit the purpose of a scientific article the purposes of the choices made for the experimental tests should be discussed.

Reply:

(1)We calculated extensional stiffness coefficients, listed in Table 2, which show the design of layups.

(2)We added explanation in section 2.1:

The other helicoidal layups were mixed with more 0° plies in addition the origin 0° plies of the LTA laminate to maintain the continuity.

The extensional stiffness coefficients and bending stiffness coefficients were calculated, see Table 2. Specimen H1, H2 and H3 are QI laminates according to the extensional stiffness coefficients. More 0° plies increase  for specimen H4, H5 and H6, while CP-73 has the maximum  due to possessing the most 0° plies.

  1. The results reported in Figure 6 to Figure 9 need to be commented on and discussed.

Reply:

1)Figure 6 is mentioned in Section 4.1.

2)Figure 7 and 8 are used to discuss the damage characteristics in Section 4.2.

3)Figure 9 is commented in section 4.3 now:

When observed perpendicularly to the plane of the laminate, the out profile of the loci of the maximum  reflects the shape of C-scan image show in Figure 9. The sector delaminated region is obvious for H1 and H2, and the protruding part on C-scan image of H4, H5, and H6 due to the multiple 0° ply is clearly shown by .

  1. Is it possible to justify the evidence discussed on page 11, line 284 about the different trends of the stiffness K_0 and K_d with the twist angle?

Re:

According to Eq.(1), the kb is determined with values of D11, D12, D66 and D22. If the D12 and D16 is close for two specimens, the closer D11 and D22 yields the higher, so k0 of H1, H2 and H3 increase gradually. K0 is related to both K0 and the damage extent, therefore is complicate to demonstrate it.

4.To give an idea about the accuracy of the stiffness value given by equation (1), the percentage difference K_b - K_0 could be added to Figure 14.

Re:The residual stiffness kd was expressed in percentage and added in Figure 14.

5.Which data from the FE analysis was used to obtain the results in Figure 18?

Re:We added: The material properties assigned to each lamina are reported in Table 1.

6.Regarding the FE model, some aspects need to be clarified

  1. Give details about the contact pairs between the plate and the indenter.

Re: We added: A frictionless hard contact was created between upper surface of the composite laminate and the outer surface of indenter.

  1. How did you evaluate the mesh convergence of the model?

Re: For the result we employed in the paper is only the location of maximum shear stress rather than the exact value, we only conducted a simple convergence analysis with two set of mesh with different size. We added:

The mesh was refined in the central region of the laminate, with a minimum size of 0.2 mm after mesh convergence analysis. Analysis showed that large mesh sizes, such as 0.4 mm, re-sult in location discrepancies in the maximum shear stress at each layer, especially if the de-viation angle is small. Integration points for adjacent layers may overlap, resulting in ap-proximations and reduced accuracy, with a maximum stress point to a distance error of up to 19%.

  1. Considering that the simulation regards a contact problem, the analysis should not be quasi-static but explicit in order to take into account the dynamic effects that in this case cannot be neglected.

Re: Dynamic response of laminate during low velocity impact is approximately quasi-static according to the study of Olsson [40]; and we primarily intended to investigate the shear stress distribution in the composite laminate mixing with 0° orientation plies as indicated by reference [12], rather than the damage and response, so the quasi-static solver was adapted in order to save computer time.

  1. The Von Mises stress is not adequate to measure the equivalent stress state in composite laminates. Another contour plot should be reported.

Re: We have made the appropriate changes to Figure 18 and updated the model. We hope that these updates address your concerns and thank you for your helpful comments.

7.Regarding the FE buckling analysis in section 4.4, how did you consider the damage related to the impact? Were the elastic properties of the plates reduced? If the damage was not considered these analyses cannot be correlated with compression after impact tests.

Re: The intact model was used in the buckling analysis. The buckling load is used to demonstrate how to choose the materials for normal usage.

8.Carefully check the paper to correct typos and misspells.

Re: We checked and corrected some misspells, thanks.

9.The references should be enlarged with more papers regarding the bending stiffness of laminates, such as: https://doi.org/10.1016/j.compstruct.2013.12.011; https://doi.org/10.1016/j.tws.2022.110374; https://doi.org/10.1016/j.apm.2022.06.012

Re: We supplied papers in the 8th paragraph of the introduction.

Reviewer 3 Report

Overall, this work provide meaningful insights into the effects of the additional 0°C plies on mechanical properties of helicoidal laminates. The findings and conclusion were supported by scientific discussions. However, several sections needs to be clarified and elaborated, and data visualization can be improved to be reader-friendly. Based on these assessments, I recommend publishing this work after the following points are addressed.

1. In the abstract, there should be an introductory sentence introducing the  key topic, for example, on the effects of the ply orientation of polymer composites filled with helicoidal laminates on their mechanical properties for engineering structural applications. Also, the based polymer (e.g., epoxy) should be clearly indicated.

2. In the introduction, it will be helpful for the reader, especially non-specialists who are not familiar with this kind of structure, if the first paragraph elaborates basic background of the Bouligand structure. The paragraphs briefly introduces many topics, but it seems that they are not fully extended to be comprehensive enough for new readers to easily understand their basic features, unique and advantageous characteristics, and importance of the Bouligand structure for focused applications. Also, the following presented ideas should be extended.

2.1. Lines 23-24: It is unclear why FRP laminates are widely used in engineering structures with high safety requirements.

2.2. Lines 30-31: It is unclear about advantages of Bouligand and why this structure is focused on this study. To improve the coherence, the first paragraph is expected to provide a general idea of the structure before presenting detailed information in the next paragraph.

2.3. Lines 36-37, “Each Bouligand unit (Figure 1) contains a helicoidal layup with very 36 small pitch angles (from 6.2 to 1.6) [8,9] for a rotation of 180° inside each unit.” This background should be presented earlier in the first paragraph.

3. Several ideas presented are too general and should be fully extended.

3.1. Lines 48-19, “Significant progress has been made in understanding the Bouligand structures and studies have been conducted toward engineering solutions.”: This statement  is too broad. It will be good if it is rewritten to narrow down to specific aspects presented in the paragraphs.

3.2. Line 78: It is unclear why the three-point-bending test is important and focused on to represent the material's mechanical properties associated to the focused application.

3.3. Lines 110-112, “Therefore, the proportion of 0°plies is constant if the helicoidal structure has a uniform pitch angle; hence, more 0°plies should be mixed into the laminate, to increase its CAI strength in one orientation”: The statement should be rewritten purposefully to show how it is related to objectives or studies being done in this work.

4. In the last three paragraphs of the introduction section should be rewritten to improve the coherence between THE focused problems (research gaps to be filled) and the objective(s) and the proposed investigation being done. Also, the problems should be better emphasized.

5. The Result and Discussions sections should be combined to improve coherence of logical discussions.

6. It will be helpful if the manuscript also include a discussion on how the findings provide guidance on directions and potential approaches to the development of high-performance engineering structure materials. Also, What are limitations of the findings and how to improve or extend the understanding in the future.

7. In the conclusion section, the following information should be emphasized.

7.1 The material should be clearly indicated.

7.2 Lines 439-440: The comparative level (by how many percent) should be indicated when reporting a comparison or a change.

7.3 Lines 447-450, “The specimen with the plies of 10° and 20° pitch angles and continuous 0° plies (H5), yielded the highest CAI strength despite more damage than the one with more plies in 0° and a higher compressive modulus (H6). The critical buckling load of H5 was also higher than that of H6.”: Scientific reasons behind the following conclusions should also be emphasized.

8. The following parts should be addressed to improve clarity.

8.1. Line 88, in the number “7” after “Grunenfelder et al.” refer to reference [7]?

8.2. Lines 158-164 are repetitive to the previous paragraph.

8.3. Line 178: The full word of “CT images” should be indicated.

8.4. In line 241, Eq. (1) is confusing. Adding brackets will make it clearer. Also, “m,n odd” should be moved to a paragraph.

8.5. Reference(s) for equations (e.g., equations (2), (3), (4), (5)) should be cited properly. Are equations (4) and (5) from Ref. [10]?

8.6. The definition of kI′, kII′, kIII′ and also KI, KII, and KIII in equations (4) and (5) are unclear. Line 352 describes that kI, kII and kIII are the local stress intensity factors along the crack front, but the is not description of the prime symbol (′). How are kI′, kII′ and kIII′ as well as KI, KII and KIII different from kI, kII and kIII, respectively?

8.7. New line and paragraph format: Lines 345-349, after (3) “In the context…”; Lines 351-353, after (4) “where v is the Poisson ratio…”; and Line 354-366, after (5) “Assuming Z=0.5…”.

9. Figures and data visualization must be improved.

9.1. There are too many figures. Some of them, for example, Fig. 3-5, Fig. 7-10, Fig. 13-14, Fig. 15-18, Fig. 19 and 21 can be combined with subfigures (a), (b), (c), etc. For Fig. 7-10, the subfigures can be organized to rows of samples and columns of the impact site, distal surface, ultrasonic C-scan image, and side cross-sectional view of the laminate’s central area.

9.2. I suggest converting the data visualization format of plots in Fig. 11 to scatter plots (for example, Fig. 11a, Kb (horizontal) vs Peak load (vertical)) with data points presented using unique symbols/colors for the samples with labels. This will be better representing data visualization that easier to understand. Similarly, for Fig. 12(a) and 12(b), a scatter plot of Kb (horizontal) VS CAI strength (vertical) should be moved to become Fig. 11(e) and also a scatter plot of Kb (horizontal) VS Compressive modulus (vertical) for Fig. 11(f).

9.3. Fig. 12 (c) and (d) should be combined into a multiple column chart (similar to Fig. 14), showing both the CAI strength and compressive modulus for each sample.

9.4. In Fig. 19, the numbers in the scale are eligible. The scale should be presented outside the image.

9.5. Fig. 20  resize to fit in 1 page.

Author Response

  1. In the abstract, there should be an introductory sentence introducing the  key topic, for example, on the effects of the ply orientation of polymer composites filled with helicoidal laminates on their mechanical properties for engineering structural applications. Also, the based polymer (e.g., epoxy) should be clearly indicated.

Re: We added:

The helicoidal laminate is a kind of nature inspired fiber reinforced polymer, and the ply orientation of affects their mechanical properties for engineering structural applications.

  1. In the introduction, it will be helpful for the reader, especially non-specialists who are not familiar with this kind of structure, if the first paragraph elaborates basic background of the Bouligand structure. The paragraphs briefly introduces many topics, but it seems that they are not fully extended to be comprehensive enough for new readers to easily understand their basic features, unique and advantageous characteristics, and importance of the Bouligand structure for focused applications. Also, the following presented ideas should be extended.

2.1. Lines 23-24: It is unclear why FRP laminates are widely used in engineering structures with high safety requirements.

        Re: We re-written:

Composite materials such as fiber reinforced polymer (FRP) laminates are widely used in engineering structures with lightweight and high safety requirements, such as aerospace structures.

2.2. Lines 30-31: It is unclear about advantages of Bouligand and why this structure is focused on this study. To improve the coherence, the first paragraph is expected to provide a general idea of the structure before presenting detailed information in the next paragraph.

        Re: We added in Paragraph one.

A helicoidal composite with uniform inter-ply angles quasi-isotropic (QI), but some laminates are required to have directionally advantageous performance in stiffness or strength, such the wing skin subjected to single directional bending. The helicoidal composites mixed with additional 0° plies were investigated in this study on its impact performance and residual strength under compression after impact (CAI).

2.3. Lines 36-37, “Each Bouligand unit (Figure 1) contains a helicoidal layup with very 36 small pitch angles (from 6.2 to 1.6) [8,9] for a rotation of 180° inside each unit.” This background should be presented earlier in the first paragraph.

Re: we revised the sentence to make its clearly:

Each Bouligand unit of the dactyl appendage (Figure 1) contains a helicoidal layup with very small pitch angles (from 6.2° to 1.6°) [8,9] for a rotation of 180° inside each unit.

  1. Several ideas presented are too general and should be fully extended.

3.1. Lines 48-19, “Significant progress has been made in understanding the Bouligand structures and studies have been conducted toward engineering solutions.”: This statement  is too broad. It will be good if it is rewritten to narrow down to specific aspects presented in the paragraphs.

Re: thanks, we revised it:

Significant progress has been made in understanding the damage growth in Bouligand structures and studies have been conducted toward engineering materials.

3.2. Line 78: It is unclear why the three-point-bending test is important and focused on to represent the material's mechanical properties associated to the focused application.

        Re: we revised this sentence:

According to the three-point-bending test, the resistance to the growth of cracks at the back face of the helicoidal laminate is less when the twist angle is small [14], so the in-ter-ply angle on the back face should be larger

3.3. Lines 110-112, “Therefore, the proportion of 0°plies is constant if the helicoidal structure has a uniform pitch angle; hence, more 0°plies should be mixed into the laminate, to increase its CAI strength in one orientation”: The statement should be rewritten purposefully to show how it is related to objectives or studies being done in this work.

        Re: we revised:

However, the proportion of 0°plies is constant if the helicoidal structure has a uniform pitch angle; hence, more 0°plies should be mixed into the laminate, to increase its stiffness and strength in 0° orientation.

  1. In the last three paragraphs of the introduction section should be rewritten to improve the coherence between THE focused problems (research gaps to be filled) and the objective(s) and the proposed investigation being done. Also, the problems should be better emphasized.

        Re: we revised some sentences in the last three paragraphs to depict the relationship between the test method, models and research objectives.

  1. The Result and Discussions sections should be combined to improve coherence of logical discussions.

        Re: we combined them and revised some sentences.

  1. It will be helpful if the manuscript also include a discussion on how the findings provide guidance on directions and potential approaches to the development of high-performance engineering structure materials. Also, What are limitations of the findings and how to improve or extend the understanding in the future.

        Re:

1) In section 3.3, we added: The cracks in the 0° direction form a relatively large projection area in the C-scan image, so the number of continues 0° plies on the impact surface should be limited.

2) In conclusion section we added:

The first ply in all specimens is of 0°, therefore, method should be proposed to evalu-ate the effect of the first ply direction on its performance. Non-symmetric and non-uniform layup with low coupling stiffness could be optimized to obtain better performance.

  1. In the conclusion section, the following information should be emphasized.

7.1 The material should be clearly indicated.

Re: we revised.

7.2 Lines 439-440: The comparative level (by how many percent) should be indicated when reporting a comparison or a change.

Re: Thanks. This description is not quit right, therefore we delete it.

7.3 Lines 447-450, “The specimen with the plies of 10° and 20° pitch angles and continuous 0° plies (H5), yielded the highest CAI strength despite more damage than the one with more plies in 0° and a higher compressive modulus (H6). The critical buckling load of H5 was also higher than that of H6.”: Scientific reasons behind the following conclusions should also be emphasized.

        Re: we added: because a small pitch angle (5°) should be avoided due to the weak constraint be-tween adjacent plies.

  1. The following parts should be addressed to improve clarity.

8.1. Line 88, in the number “7” after “Grunenfelder et al.” refer to reference [7]?

        Re: Thank, we revised it.

8.2. Lines 158-164 are repetitive to the previous paragraph.

        Re: Thank, we revised it.

8.3. Line 178: The full word of “CT images” should be indicated.

        Re: Thank, we revised it.

8.4. In line 241, Eq. (1) is confusing. Adding brackets will make it clearer. Also, “m,n odd” should be moved to a paragraph.

Re: Thank, we revised it.

8.5. Reference(s) for equations (e.g., equations (2), (3), (4), (5)) should be cited properly. Are equations (4) and (5) from Ref. [10]?

Re: Yes, we revised it.

8.6. The definition of kI′, kII′, kIII′ and also KI, KII, and KIII in equations (4) and (5) are unclear. Line 352 describes that kI, kII and kIII are the local stress intensity factors along the crack front, but the is not description of the prime symbol (′). How are kI′, kII′ and kIII′ as well as KI, KII and KIII different from kI, kII and kIII, respectively?

Re: The prime symbol is missing. We revised it.

8.7. New line and paragraph format: Lines 345-349, after (3) “In the context…”; Lines 351-353, after (4) “where v is the Poisson ratio…”; and Line 354-366, after (5) “Assuming Z=0.5…”.

        Re: we are not sure.

  1. Figures and data visualization must be improved.

9.1. There are too many figures. Some of them, for example, Fig. 3-5, Fig. 7-10, Fig. 13-14, Fig. 15-18, Fig. 19 and 21 can be combined with subfigures (a), (b), (c), etc. For Fig. 7-10, the subfigures can be organized to rows of samples and columns of the impact site, distal surface, ultrasonic C-scan image, and side cross-sectional view of the laminate’s central area.

Re: As per your suggestion, we have made changes to the figures in our manuscript to improve the clarity of our findings. Specifically, we have combined Fig. 3-5 and Fig. 15-16 with subfigures (a), (b), (c), and for Fig. 7-9, we have re-organized the subfigures to rows of samples and columns of the impact site, distal surface as you suggested, while the Fig.9 ultrasonic C-scan image, and Fig.10 side cross-sectional view of the laminate’s central area remain unaltered for better convenience in performing figure-based analyses in the text. The original Fig. 17 and 21 in the latest version of the manuscript were removed and we have updated the corresponding content and figure numbering in the manuscript accordingly.

9.2. I suggest converting the data visualization format of plots in Fig. 11 to scatter plots (for example, Fig. 11a, Kb (horizontal) vs Peak load (vertical)) with data points presented using unique symbols/colors for the samples with labels. This will be better representing data visualization that easier to understand. Similarly, for Fig. 12(a) and 12(b), a scatter plot of Kb (horizontal) VS CAI strength (vertical) should be moved to become Fig. 11(e) and also a scatter plot of Kb (horizontal) VS Compressive modulus (vertical) for Fig. 11(f).

        Re: this is a good suggestion, but the order of specimen will be change if the data was plotted in Kb (horizontal) vs Peak load (vertical), so we didn’t revised it. Figure 12 (Fig.9 now) is used to emphasize CAI strength, so we didn’t revise it.

9.3. Fig. 12 (c) and (d) should be combined into a multiple column chart (similar to Fig. 14), showing both the CAI strength and compressive modulus for each sample.

        Re: In Fig.14(Fig.11 now), k0 and kd have the same order of magnitude, so multiple column chart is used; but Fig 12 (Fig.9 now) is used to demonstrate CAI, and other parameter (D, kb) have the different order of magnitude of CAI strength 

9.4. In Fig. 19, the numbers in the scale are eligible. The scale should be presented outside the image.

Re: We have made the necessary changes to address your suggestion regarding Fig. 19 in our manuscript. The scale has been moved outside the image to improve its legibility. Additionally, we have updated the figure labeling, and this figure now corresponds to the most recent version of Fig. 14.

9.5. Fig. 20 resize to fit in 1 page.

Re: we have reorganized the layout of the text and adjusted the position of the image to fit it all onto one page, and this figure is now labeled as Fig.15.

Round 2

Reviewer 2 Report

The suggested corrections have been considered.

As far as I am concerned, the paper can be published.